# Dysbiosis and Gastrointestinal Surgery: Current Insights and Future Research

**DOI:** 10.3390/biomedicines10102532

**Published:** 2022-10-10

**Authors:** Giulia Gibiino, Cecilia Binda, Ludovica Cristofaro, Monica Sbrancia, Chiara Coluccio, Chiara Petraroli, Carlo Felix Maria Jung, Alessandro Cucchetti, Davide Cavaliere, Giorgio Ercolani, Vittorio Sambri, Carlo Fabbri

**Affiliations:** 1Gastroenterology and Digestive Endoscopy Unit, Forlì-Cesena Hospitals, Ausl Romagna, 47121 Forlì-Cesena, Italy; 2Department of Medical and Surgical Sciences—DIMEC, Alma Mater Studiorum—University of Bologna, 40138 Bologna, Italy; 3General and Oncologic Surgery, Morgagni—Pierantoni Hospital, AUSL Romagna, 47121 Forlì-Cesena, Italy; 4Microbiology Unit, Hub Laboratory, AUSL della Romagna, 47121 Forlì-Cesena, Italy

**Keywords:** antibiotic prophylaxis, mechanical bowel preparation, bariatric surgery, obesity and microbiota

## Abstract

Surgery of the gastrointestinal tract can result in deep changes among the gut commensals in terms of abundance, function and health consequences. Elective colorectal surgery can occur for neoplastic or inflammatory bowel disease; in these settings, microbiota imbalance is described as a preoperative condition, and it is linked to post-operative complications, as well. The study of bariatric patients led to several insights into the role of gut microbiota in obesity and after major surgical injuries. Preoperative dysbiosis and post-surgical microbiota reassessment are still poorly understood, and they could become a key part of preventing post-surgical complications. In the current review, we outline the most recent literature regarding agents and molecular pathways involved in pre- and post-operative dysbiosis in patients undergoing gastrointestinal surgery. Defining the standard method for microbiota assessment in these patients could set up the future approach and clinical practice.

## 1. Introduction

The gut microbiota (GM) consists of a complex ecosystem that is responsible for our health and characterizes us in a unique way, like a fingerprint [1]. Through metabolites and complex molecular pathways, GM interacts with multiple organs and diseases within a multidirectional network [2]. It is therefore to be expected that surgery on the digestive system may somehow disrupt the balance of the intestinal barrier with short- and long-term consequences. On the other hand, many surgically treated diseases are originally characterized by dysbiosis. It thus remains to be understood which is the cause and which the effect [3].

Certainly, advanced techniques for studying the microbiota, which have begun to spread in recent years, will open up new perspectives on dysbiosis factors predictive of surgical complications and poorer prognosis. Even in the surgical world, a role for the microbiome as an organ is now accepted, and its potential to be saved pre- and post-operatively is understood. The intent of this descriptive review is to summarize the latest evidence and research trends in the relationship between microbiota and digestive surgery. 

## 2. Methods

We selected articles discussing the association of gastrointestinal surgery and microbiota. In particular, we chose articles published in the last twenty years, focusing on the latest scientific evidence. The studies were mostly European and American, with a minority performed on the eastern population. We developed a non-systematic review article using the following electronic sources: PubMed, EMBASE, Google Scholar, Ovid, MEDLINE, Scopus, Cochrane controlled trials register, and Web of Science. We used the following single combined search terms: “Gastrointestinal Surgery AND Gut microbiota”, “GI Surgery AND microbiota”, “Post-surgery AND Gut microbiome”, “Post-surgical infections AND microbiota”, “Bariatric Surgery AND microbiota”. We examined all the articles reporting humans’ related data (inclusion criteria) excluding works with not available full text, not in English language, book chapters, abstracts, and articles published before 1990 (exclusion criteria). Finally, we evaluated supplementary references among articles evaluated in the first search round.

## 3. Preoperative Management for Elective Colorectal Surgery: The Importance of Preserving the “Good” Resident Microbiota

Evidence from recent years shows that preparing the gut for surgery can affect the health of the microbiota, and thus the long-term prognosis. There has been a shift from the idea of sterile surgery to the realization that the “good” microbiota must be preserved. The issue is to avoid all the potential pathogens as much as possible. 

Adoption of purgative cleansing, oral and intravenous antibiotics and application of topical antiseptic solutions is part of the routine practice usually adopted nowadays. The scientific community is now understanding the possible effects on gut microbiota. The administration of osmotic agents, such as polyethylene glycol (PEG) which is a metabolically inert isotonic laxative, is the most diffuse way to obtain adequate cleansing in the absence of presumed relevant consequences and side effects. However, there is still debate on the effective advantages of this practice, and moreover, preserving gut fecal contents seems to lead to better outcomes [4]. Mechanical bowel preparation impacts the complex intestinal barrier by interacting with endoluminal commensals and epithelial components. A recent meta-analysis conducted on eight studies and 1065 patients showed that bowel preparation does not lower the risk of colorectal anastomotic leakages [5]. Beyond the evidence of microbiota variation during diagnostic and operative endoscopy cleaning, important data are emerging about the consequences for those undergoing elective colon-rectal surgery. A recent Cochrane metanalysis showed that there is no statistically significant evidence that patients benefit from mechanical bowel preparation, nor the use of rectal enemas. In colonic surgery, bowel cleansing can be safely omitted and induces no lower complication rate [6]. In 2010, Watanabe et al. first published a study based on a Japanese population evaluating the impact of bowel preparation in this specific setting [7]. They reported decreased populations of *Bifidobacteria*, *Clostridium coccoides*, *Clostridium leptum*, *Enterobacteriaceae*, and *Lactobacillus* postoperatively, with no change in *Enterococcus* and *Staphylococcus* populations. This alteration with reduced levels of short-chain fatty acids (SCFAs) could result in impairment of the intestinal barrier, thus leading to bacterial translocation and possible infectious complications. Similar results were shown in another Chinese RCT with significant reduction in the whole bacterial abundance, especially regarding *Bacteroides* and *Peptostreptococcus*, and bacillus/coccus ratio [8]. Further data were analyzed considering the days required for bowel preparation. Sixty patients undergoing colorectal resection were enrolled in Nanfang Hospital from March 2010 to March 2011 and were randomly assigned to receive 3 days vs. 1 day of bowel preparation. They observed postoperative changes in both groups; the postoperative population of *Bifidobacterium* and *Lactobacillus* showed a significant decrease, which was more significant in the group undergoing fast 1-day preparation, while E. coli and Staphylococcus were higher than the preoperative level, which was more significant in the 3-day preparation. Significant results emerged in terms of postoperative infections with lower incidence in case of fast preparation [9].

The adoption of antibiotic prophylaxis is another cornerstone of the pre-operative phase that could change from a microbiota preservation perspective. The lack of a targeting approach to pathogens still makes it a debated point despite the evidence in preventing major infections. 

The linkage between antibiotic administration and changes in gut microbiota diversity and consequent antibiotic-associated diarrhea is a concept learned long ago [10]. The mechanisms of interaction of antibiotic topical or systemic therapy preoperatively are not yet well understood and could lead to a prevalence of microbes responsible for short- and long-term post-operative consequences; surprisingly, it appears that antibiotic therapy does not lead to a significant risk of *C. difficile* colitis [11]. 

It is clear that particular attention was given to the possible combined effect of intestinal mechanical preparation and antibiotic prophylaxis. Kiran et al. [12] conducted a comparative study in 2015 for 8442 patients undergoing elective colorectal resection, evaluating the impact of preoperative osmotic preparation and antibiotics, osmotic preparation alone and no bowel preparation in terms of outcomes, particularly anastomotic leak, surgical site infection (SSI) and ileus. On multivariable analysis, mechanical bowel preparation with antibiotics, but not without, was independently associated with reduced anastomotic leak, surgical site infections and postoperative ileus. These data were in accordance with subsequent results from a large metanalysis [13]. However, the more recent MOBILE trial challenged the relative evidence that the combination of osmotic preparation and antibiotic together lead to better outcomes. They included 396 patients undergoing elective colectomy, randomly assigned to combined preparation (196 patients) and no bowel preparation (200), and no significant reduction of surgical site infections and overall morbidity after combined preparation [14]. This raised the question of assessing the benefits by combining the two factors, and the ORALEV study recently confirmed that the administration of oral antibiotics as prophylaxis the day before colon surgery significantly reduces the incidence of surgical-site infections without mechanical bowel preparation, since mechanical bowel preparation was used in both groups [15]. 

Interesting results are expected from the ongoing MOBILE2 trial, which compares mechanical and antibiotic bowel preparation versus bowel preparation alone in patients undergoing rectal surgery. The aim of the study is to examine if oral antibiotics reduce the overall complications, SSIs or anastomotic leakages after rectal surgery, and also if there are any adverse effects related to the oral antibiotics [16]. 

These studies suggest that there is ongoing interest in the best practice to prevent dangerous infections while preserving our “good” commensals, and there is still a long way to go to achieve consensual international recommendations. 

## 4. Which Microbes and Which Molecular Pathways after Gastrointestinal Surgery

The first systematic review assessing pre- and post-surgical microbiome changes after gastrointestinal surgery was published in 2021. They included 14 studies, all reporting post-surgical changes in the microbiome profiles. In 9 of the 14 studies, the prevalence of specific bacteria had significantly changed after surgery. Improved outcome was associated with higher levels of beneficial bacteria and greater microbiome diversity post-surgery [17]. The key recommendations reported by the authors are the development of a standardized protocol for microbiome sampling methods, choice of sample sites within the GIT, analysis methodology and time frames of testing. 

Eight studies reported inflammatory bowel disease (IBD) surgical approach and its consequences [18,19,20,21,22,23,24,25,26]. Specific changes reported enhance the idea that surgery can somehow normalize a state of dysbiosis that characterizes these diseases. For Crohn’s disease, great variability is observed in the case of ileocolic anastomosis or ileostomy. Interesting features have been found in patients who develop post-surgical disease recurrence, with higher counts of nitrate-reducing bacteria such as *E. coli* 3 months after surgery. Recurrence was associated with increased levels of facultative anaerobes and nitrate reducers (*Streptococcaceae* and *Enterobacteriaceae*) [22] rather than anaerobic spore-forming bacteria such as *C. coccoides*, and other potentially beneficial *Clostridiales* such as *F. prausnitzii*, which were increased in patients in remission [23,24].

Four studies, two European and two asian [27,28,29,30], assessed microbiome changes following colorectal surgery, giving attention to some key players: the decrease in *Firmicutes*, *Bacteroides*, *Clostridium leptum* (*Firmicutes*), *Bifidobacterium* (*Actinobacteria), Atopobium cluster (Actinobacteria), Prevotella (Bacteroidetes)* and *Escherichia Coli (Proteobacteria)*; conversely, an increase was shown for *Enterobacteriaceae (Proteobacteria)*, *Clostridia (Firmicutes)* and Gram-negative anaerobes (including *Bacteroides, Fusobacteria* and *Pseudomonas*). There were less marked associations with microbial changes compared to the IBD population. In one study [28], surgical site infections within 1 week of surgery were associated with increased numbers of facultative anaerobes and nitrate reducers such as the *Proteobacteria Pseudomonas* and *Enterobacter*, *Bacteriodes* spp and *Firmicutes Staphylococcus*, *Enterococcus* and *Klebsiella*. Another study [31] found a similar trend towards reduced post-surgical infectious complications in the group who had decreased *Enterobacteriaceae*, *Staphylococcus* and *Pseudomonas* and an increase in *Bifidobacterium* within 2 weeks after surgery.

A more recent study by Kong et al. [32] analyzed the compositional shifts of the intestinal microbiota in fecal samples from 43 CRC patients undergoing radical surgery through 16S rRNA amplicon sequencing preoperatively compared to postoperatively. After CRC surgery, they observed a reduced ratio of *Bacteroidetes/Firmicutes*; the numbers of beneficial obligate anaerobes, including *Bacteroides*, *Bifidobacterium*, *Faecalibacterium*, *Parabacteroides* and *Prevotella*, were also reduced postoperatively. Moreover, radical surgery not only removed the tumor-associated lesions, but also eliminated well-known tumor-associated bacteria including *Enterococcus* and *Fusobacterium*, possibly aiding in recurrence prevention. Furthermore, butyrate-producing bacteria (*Bacillus*, *Bilophila*, *Barnesiella*) were decreased, whereas conditional pathogens, including *Escherichia-Shigella, Enterobacteriaceae* and *Streptococcus*, were enhanced. These results are contrary to another Chinese study that revealed a decrease in *Escherichia-Shigella* and an increase in *Enterococcus* and *Parabacteroides* [33]. Thus, these perturbations of intestinal microbiota after CRC surgery could promote adverse inflammatory outcomes in these patients. A further systematic review was recently conducted, which included 6 randomized controlled trials and 27 prospective cohort studies, reporting a total of 968 patients. Gastrointestinal surgery was associated with an increase in α diversity and a shift in β diversity postoperatively. Multiple bacterial taxa were identified to consistently trend toward an increase or decrease postoperatively. A difference in microbiota across geographic provenance was also observed. There was a distinct lack of studies showing correlation with clinical outcomes or performing microbiome functional analysis. Furthermore, there was a lack of standardization in sampling, analytical methodology and reporting [34]. 

### 4.1. Microbiota Involvement in Anastomotic Leakage and Surgical Site Infections

The process of tissue repair begins immediately after intestinal resection and anastomosis. Some evidence in animal models, such as germ-free mice, showed that the absence of commensals could delay physiological repair processes [35]. AL occurrence was usually linked to poor surgical technique-related factors, including ischemia, increased suture tension, device deployment, suture type or placement and stapling method. However, recent vast improvements in surgical technology did not lead to the expected better outcomes [36].

The available literature supports an important role for intestinal microbiota in anastomotic healing and susceptibility to anastomotic leakage (AL). All this knowledge mainly derives from animal studies conducted in rodents [37,38,39]. Butyrate-producing bacteria assist in epithelial repair, while specific collagenolytic pathogens, such as *E. faecalis* and *P. aeruginosa*, have been involved in AL pathophysiology. However, their presence should be added to other factors to develop AL. 

A recent study assessed the efficacy of bioresorbable sheath (C-Seal) in preventing colorectal AL in 123 patients undergoing colorectal resection. The C-seal was introduced to help reduce the risk of anastomotic leakage. It comprises an intraluminal biodegradable soft sheath that is stapled to the colorectal anastomosis [40,41]. They collected mucosa-associated microbiota from the stapled colorectal “donut” and performed a 16s MiSeq sequencing. Results showed that in non-C-seal population AL was related to reduced microbial diversity and increased abundance of *Bacteroidaceae* and *Lachnospiraceae* compared to patients without AL. AL was not associated with intestinal microbiota in C-seal patients who presented higher AL rates. Although C-seal placement did not reduce AL occurrence, it obliterated the association between intestinal microbiota and AL [42,43]. 

A more recent study compared preoperative microbiota composition in patients eligible for CRC surgery. They showed that patients developing AL had increased dysbiosis-related bacteria, as Acinetobacter Iwo and Hafina alvei; low abundance of protective bacteria *Barnesiella intestihominis* and *Faecalibacterium prausnitzii* were reported as well [44]. A similar preoperative assessment was conducted by Liu Y et al. [45].

Based on these data, there is an emerging concept on the possibility of preoperative microbiota screening predictive for AL development; furthermore, a preoperative modulation could be applied to prevent AL and surgical site infections (SSIs). 

In addition to AL, the problem of SSIs is emerging as a possible correlate of dysbiosis status. The infectious agents involved have long been reported, but only recently has microbiota analysis been proposed to interpret these data. Ohigashi et al. [28] is currently one of the first studies reporting fecal microbiota with SSI-related bacteria in CRC patients after surgery. SSIs occurred in 6 out of 81 patients, and the causative bacteria were identified to be *S. aureus, P. aeruginosa* and *Enterococcus* spp., which were also enriched in the analysis of fecal samples after CRC surgery. The MIRACLe protocol was proposed to evaluate the effects of a novel perioperative treatment for the implementation of the gut microbiota, to prevent anastomotic fistula and leakage (AL) in patients undergoing laparoscopic colorectal resections for cancer. This study was conducted in a single Italian center, including 60 patients undergoing elective colorectal surgery and receiving a novel perioperative preparation following the MIRACLe (Microbiota Implementation to Reduce Anastomotic Colorectal Leaks) protocol (oral antibiotics, mechanical bowel preparation and perioperative probiotics), compared to a group of 500 patients (control group), who received a standard ERAS protocol. In the MIRACLe Group, only one anastomotic leak was registered. The MIRACLe group showed encouraging results, with significant lower incidence of AL, surgical site infections, reoperations and post-operative mortality [46]. Another protocol has been recently proposed to assess whether a presurgical nutritional intervention, based on a high-fiber diet rich in polyunsaturated fatty acids (PUFAs), can reduce disturbances of the gut microbiota composition and, consequently, the rate of post-surgical complications in patients with CRC. This is expected to be the first study protocol evaluating the impact of a presurgical nutritional intervention based on high-fiber intake with high levels of PUFAs on altered gut microbiota and its relationship with post-surgical complications, anastomotic leaks and site infections. The ongoing METABIOTE trial is an observational prospective cohort study which aims to define the predictive value of prognostic markers, including gut microbiota, sarcopenia, metabolic syndrome and obesity. All consecutive patients with a non-metastatic sporadic CRC scheduled for surgery at the Digestive Surgery department at the University Hospital of Clermont-Ferrand (France) are systematically asked to participate in the study, aiming at inclusion of 300 patients from 2019 to 2021. The primary outcome is the 5-year overall survival (OS). Other outcomes are 5-year CRC-related OS, 5-year disease-free survival, 30-day postoperative morbidity, 90-day postoperative mortality and length of hospital stay [47].

### 4.2. Long-Term Microbiota Modifications after Gastrointestinal Surgery

On the basis of the above, a change in the microbiota affects the immediate post-operative phases, but it is expected that it may also persist in the long term. In this respect, it is the oncology field that has been most studied, in order to identify a prognostically favorable microbial profile.

The first bacterium studied in CRC patients in relation to oncological outcomes is Fusobacterium nucleatum (*F. nucleatum*), which is known for its role in tumorigenesis. Bacteroides fragilis (*B. fragilis*) was then correlated as well with reduced disease-free and overall survival, especially enterotoxigenic B. fragilis (ETBF) was related to CRC progression [48]. Another bacterial genus of particular interest concerning CRC outcomes is Bifidobacterium, with its abundance having been reduced in CRC patients compared with healthy individuals, as reported by a cohort study of 1313 CRC patients. They reported the detection of *Bifidobacterium* in 30% of tumors with no statistically significant association between its abundance and survival outcomes or molecular and clinicopathological features of CRC. [49] 

After individual key players, the concept of co-abundance groups (CAGs) was introduced. Flemer et al. [50] provided an analysis based on CAGs of various bacteria, and they reported that *Bacteroidetes*, *Prevotella* and pathogen CAGs were correlated with longer survival in CRC patients. 

A Chinese group [51] was the first to evaluate the long-term metabolic status and microbial composition of CRC patients after curative surgery. They showed that metabolic syndrome was more prevalent among CRC patients after right hemicolectomy (RH) but not after lower anterior resection (LAR) compared to controls over a follow-up period of 5 years. The RH group also showed dysbiosis represented by lower bacterial richness and diversity as opposed to the LAR group. The RH group also showed microbiota alterations postoperatively with a reduction in the *Firmicures/Bacteroidetes* ratio, with a higher level of *Fusobacterium* and lower numbers of butyrate-producing *Faecalibacterium prausnitzii* and *Roseburia*. Abbas et al. [52] recently published an observational cohort study evaluating gut microbiota changes before and after elective oncologic colon surgery in adults previously receiving different antibiotic prophylaxis. They performed metataxonomic analysis based on sequencing of the bacterial 16S rRNA gene marker. Despite the small population of 27 patients, they observed large and significant increases in the genus *Enterococcus* between the preoperative/intraoperative samples and the postoperative sample, mostly represented by *Enterococcus faecalis*. There were also significant differences in the postoperative microbiome between patients who received standard prophylaxis and carbapenems, specifically in the family *Erysipelotrichaceae*.

Similar evidence was suggested by the Malaysian population studied by Png et al. [53]. They evaluated forty-nine fecal samples from 25 noncancer (NC) individuals and 12 CRC patients, before and 6 months after surgery, performing analysis through bacterial 16S rRNA gene sequencing. Bacterial richness and diversity were reduced, while pro-carcinogenic bacteria such as *Bacteroides fragilis* and *Odoribacter splanchnicus* were increased in CRC patients compared to the NC group. These differences were no longer observed after surgery. The comparison between the preoperative and postoperative CRC population confirmed increased abundance of probiotic bacteria after surgery. Concomitantly, bacteria associated with CRC progression were observed to have increased after surgery, implying persistent dysbiosis. The increasing use of advanced latest-generation analysis techniques will lead to further analysis of the microbial tissue characterizing this population in the long term, providing new insights.

## 5. Bariatric Surgery

### 5.1. Microbiota and Obesity

In order to consider the impact of bariatric surgery on the microbiota, it is necessary to re-evaluate the microbial balance during obesity. 

Obesity is a chronic disease that recognizes a multifactorial etiology: environmental, dietary, lifestyle, host and genetic factors contribute to its establishment [54].

In the current understanding of this disease, gut microbiota plays an accepted role in the development of obesity; in fact, it is considered, by definition, a state of dysbiosis [55].

In obesity, a reduction in microbial genetic richness and compositional and functional alterations has been observed. Dysbiosis contributes to a state of low-grade inflammation, increased body weight and fat mass, and increased risk of type 2 diabetes [56].

The microbiota of individuals with obesity is believed to have an increased capacity for energy harvest [57].

Indeed, the gut microbiota produces short-chain fatty acids; these are bacterial metabolites derived from the fermentation of otherwise indigestible oligosaccharides, dietary plant fibers and undigested proteins. Short-chain fatty acids are involved in an important part of the pathophysiology of obesity, namely stimulation of the production of satiety hormones (GLP-1 and PYY), as well as playing a role in lipid metabolism, inflammation and insulin sensitivity. 

In addition, specific microbial profiles have been observed in obesity. As widely accepted, obesity is commonly associated with an increased ratio of *Firmicutes* to *Bacteroidetes*, a dysbiotic energy-harvesting microbiome [58].

The first treatment of overweight and obesity is usually clinical management, traditionally based on lifestyle interventions [59]. Weight loss through dietary interventions alters the composition of the gut microbiota, counteracting the high *Firmicutes*/*Bacteroidetes* ratio widely reported in obesity and increasing the benefits of the phylum *Verrucomicrobia* [60], but also promoting functional changes in the microbiota and altering its derived metabolites [61,62]. When nonsurgical alternatives have failed, bariatric surgery (BS) is considered the gold standard treatment, with great performance in remediating the condition in the short and long term [63].

### 5.2. Evidence after Surgery

In the last few decades, BS has increased dramatically worldwide and now appears to be not only a treatment for weight loss but a solution to reduce cardiovascular risks and diabetes, thus leading to this surgical procedure being considered a “metabolic surgery”. 

First, bariatric surgery has a profound effect on biochemical and anthropometric parameters, improving the health status of patients in the years following surgery [64].

The known mechanisms by which bariatric surgery succeeds in achieving its therapeutic effects are: altered diet (significantly energy restricted with increased protein intake), changes in the anatomy and physiology of the gastrointestinal tract (GIT) induced by the type of procedure performed (thus, with specific changes in food digestion and absorption) and lack of adequate nutritional supplementation [65].

A link between GM change and improved body composition after surgery-induced weight loss is evidenced by animal and human fecal transplants from BS patients to mice reared without any exposure to the microorganisms. In these studies, after transplantation and over three months, mice who received microbiota from BS patients gained less fat mass than mice colonized with microbial communities from obese patients [56].

Studies using different sequencing methods have reported elevated microbial gene richness (MGR) and diversity of gut microbial populations after Roux-en-Y gastric bypass (RYGB) and vertical sleeve gastrectomy (VSG) and a shift from “obese” to “less obese” microbial structure [66].

One of the most interesting studies in this area, using high-resolution sequencing technology, has shown that bariatric surgery, including both adjustable gastric banding (AGB) and RYGB, improves MGR. In most patients it is partially restored, and most maintain low MGR (even for longer periods, such as 5 years) while losing a lot of weight. [64].

At the phylum level, bariatric surgery appears to reduce the abundance of the phylum *Firmicutes*, while *Proteobacteria* showed an opposite pattern [67].

This change in the composition and richness of the gut microbiota after surgery appears to be more profound in Roux-en-Y gastric bypass (RYGB) than in sleeve gastrectomy [68,69]. In other studies, changes in gut microbiota among patients undergoing these two types of interventions appear to be similar [70,71]. A recent study, analyzing the fecal metagenome and metabolome of patients with severe obesity after BS, demonstrated long-term improvement in GM even after a longer period (up to 4 years). [72]

Having ascertained the actual change in GM after BS and having hypothesized a possible role in improving body composition, the main point to be understood is whether there are interventions that can be used in clinical practice before BS that can enhance this change. Several studies have found that the use of probiotics has the potential to reduce SIBO and improve gastrointestinal symptoms after bariatric surgery [73,74]. Another recent study showed that *L. acidophilus* and *B. lactis* supplementation is effective in reducing swelling but without affecting the development of SIBO in the early postoperative period [75]. The use of prebiotics before BS could also improve metabolic effects, such as increased postprandial secretion of insulin, GLP-1, and PYY, which usually increases after surgery. This might be a starting point to investigate, with new studies, whether the use of prebiotics in the late postoperative period might be more effective in patients with a weak response to insulin and incretins, and thus insufficient weight loss or diabetes [76].

Another interesting aspect is the possible effect of gut microbiota on the nervous system and cognitive physiology. Evidence from animals and humans implies that the gut microbiota affects brain structure and cognitive functions [77]. A recent longitudinal study highlights the crucial role of the gut microbiota–mycobiota in human somatic and cognitive health, particularly memory, suggesting that the complex ecology of the gut–brain axis evolves dynamically to adapt body and brain physiology in response to BS [78].

## 6. Summary

The gut microbiota acts as an ecosystem with a very important role in the well-being of the entire organism. It has also been shown that the microbiota has a great influence on the healing process of an intestinal anastomosis. There is a strict relationship between GM and the surgical approach considering that surgical intervention is a curative trauma for this ecosystem. The goal of the future research should be evaluating the “good” microbes improving surgical outcomes. At the same time, pathogens are still the mainstay of mortality in this setting and should not be underestimated. Further large studies will highlight the possible microbiota profiles predictive of digestive surgery outcomes, and everyday practice will be influenced by this rapidly growing evidence. Using the increasingly available and inexpensive methods of personalized microbiota analysis, we can advance the understanding of the intestinal microbiota in health and disease and ultimately guide clinical care. In addition, new studies are needed to determine whether the use of pharmacological interventions that modify the gut microbiota can improve some post-surgical outcomes by increasing the prevalence of protective versus pathogenic microbial species.

## Data Availability

Not applicable.

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
