# Peer review of "Dysbiosis and Gastrointestinal Surgery: Current Insights and Future Research"

_biomedicines, 2022, doi:10.3390/biomedicines10102532_

Round 1
Reviewer 1 Report
The topic of gut microbiota is of a big importance because of the role it may play in a lot of pathologies. The article is a narrative review which tries to summarise the data available regarding the involvement of the microbiota in digestive surgery. In my opinion the aim of the review is to extensive, every chapter of the results/discussions may be an article itself.
Author Response
Response to Reviewer #1:
“The topic of gut microbiota is of a big importance because of the role it may play in a lot of
pathologies. The article is a narrative review which tries to summarise the data available regarding
the involvement of the microbiota in digestive surgery. In my opinion the aim of the review is to
extensive, every chapter of the results/discussions may be an article itself”.
We thank the reviewer for the comment. We have shortened the paragraph on obesity, trying to be
more concise and give a common thread to the whole manuscript.

Reviewer 2 Report
The manuscript is interesting, Authors included latest information about the effects of gastrointestinal surgery and widely discussed them.
I would like to suggest: 1. the manuscript will be more useful for the readers if a schema of methodology of choosing publications for this review will be included;
2.the geographical differences between patients should be pointed out (different diets, supplements etc.) unless the review is focused only on Europe/USA.
3.Conclusions -it is rather summary, so I am suggesting to change the title to Summary and/or add real conclusion(s).
Author Response
Response to Reviewer #2:
The manuscript is interesting, Authors included latest information about the effects of
gastrointestinal surgery and widely discussed them.
I would like to suggest:
1. the manuscript will be more useful for the readers if a schema of methodology of choosing
publications for this review will be included;
2.the geographical differences between patients should be pointed out (different diets, supplements
etc.) unless the review is focused only on Europe/USA.
3.Conclusions -it is rather summary, so I am suggesting to change the title to Summary and/or add
real conclusion(s).
We thank the reviewer and we welcome his suggestions:
1. We added a paragraph including searching methods;
2. We added details on geographical details when available. Diets and supplements were not
reported;
3. We have changed the title from conclusion to summary.

Round 2
Reviewer 1 Report
The topic issued by the authors is very large and it is very hard to make a such comprehensive review regarding the involvement of microbiota in gastrointestinal surgery. Every part o the bowel (stomach, small bowel, colon and rectum) has its particular ecosystem. The article it is well written and on a actual topic.
Maybe another title would be more appropriate.
Author Response
We thank the reviewer for the comment. We changed the title as suggested.